# Liquid-Phase Exfoliation of Graphene: An Overview on Exfoliation Media, Techniques, and Challenges

**DOI:** 10.3390/nano8110942

**Published:** 2018-11-15

**Authors:** Yanyan Xu, Huizhe Cao, Yanqin Xue, Biao Li, Weihua Cai

**Affiliations:** 1School of Architecture, Harbin Institute of Technology, Harbin 150001, China; xueyuwulai@126.com; 2School of Energy Science and Engineering, Harbin Institute of Technology, Harbin 150001, China; caiwh@hit.edu.cn; 3Key Laboratory of Cold Region Urban and Rural Human Settlement Environment Science and Technology, Ministry of Industry and Information Technology, Harbin 150001, China; 4College of Materials Science and Chemical Engineering, Harbin Engineering University, Harbin 150001, China; xueyanqin@hrbeu.edu.cn

**Keywords:** graphene, liquid-phase exfoliation, exfoliation media and techniques

## Abstract

Graphene, a two-dimensional (2D) carbon nanomaterial, has attracted worldwide attention owing to its fascinating properties. One of critical bottlenecks on some important classes of applications, such as printed electronics, conductive coatings, and composite fillers, is the lack of industrial-scale methods to produce high-quality graphene in the form of liquid suspensions, inks, or dispersions. Since 2008, when liquid-phase exfoliation (LPE) of graphene via sonication was initiated, huge progress has been made in the past decade. This review highlights the latest progress on the successful preparation of graphene in various media, including organic solvents, ionic liquids, water/polymer or surfactant solutions, and some other green dispersants. The techniques of LPE, namely sonication, high-shear mixing, and microfluidization are reviewed subsequently. Moreover, several typical devices of high-shear mixing and exfoliation mechanisms are introduced in detail. Finally, we give perspectives on future research directions for the development of green exfoliation media and efficient techniques for producing high-quality graphene. This systematic exploratory study of LPE will potentially pave the way for the scalable production of graphene, which can be also applied to produce other 2D layered materials, such as BN, MoS_2_, WS_2_, etc.

## 1. Introduction

Graphene is a new type of two-dimensional (2D) carbon nanomaterial, which exists as a single layer of carbon atoms that are tightly arranged in a honeycomb lattice. It is the basic constructional element for all other dimensions of graphitic materials (Figure 1) [1]. It can be wrapped up into zero-dimensional (0D) fullerenes, rolled into one-dimensional (1D) carbon nanotubes, or stacked into three-dimensional (3D) graphite. Since graphene was first prepared by Novoselov and Geim using micromechanical exfoliation of graphite [2], which is also called the ‘Scotch tape’ method [3], it has been fascinating to many researchers. Owing to its distinct mechanical, electronic, physical and thermal properties, widespread applications of graphene have been explored, such as transparent/flexible electrode [4,5], sensor [6], energy storage (Figure 2) [7], composites [8], etc. 

One of the critical bottlenecks on some important classes of applications, such as printed electronics, conductive coatings, and composite fillers, is the lack of industrial-scale methods to produce high-quality graphene in the form of liquid suspensions, inks, or dispersions [9,10]. At present, there are many methods for preparing graphene with different sizes, shapes, and quality. These methods can be divided into two major categories, the bottom-up method and the top-down method [11]. The most common methods adopted for graphene production shown in Figure 3, play a vital role in determining the properties of the final product [12]. Different methods offer choices suitable for specific applications (Figure 4) [9]. The bottom-up methods, such as epitaxial growth and chemical vapor deposition (CVD) [13], can produce thickness-controllable and large-size graphene. However, these substrate-based techniques are limited by restricted dimensions and high cost, so that it cannot meet the requirement for commercial application of high-quality graphene [14]. Top-down methods [15], such as reduction of graphene oxide [16] generally involve three steps: the preparation of graphite oxide (GO) via chemical oxidation [17], the exfoliation of GO using sonication to prepare the dispersions of GO, and the reduction of dispersions of GO [13]. However, doughty chemical oxidation destroys the electronic structure of graphene, which limits the application in the microelectronics, etc. LPE, a new top-down method, can obtain a stable dispersion of monolayer or few-layer defect-free graphene, which only involves the exfoliation of natural graphite via high-shear mixing or sonication [18]. Therefore, LPE is more universal compared with most of the other methods, and LPE of graphene will become a significant technique in the not-so-distant future [19,20], some Chinese companies have industrially produced graphene by LPE, as shown in Table 1. 

Since 2008, when LPE of graphite through the sonication of graphite powder in N-methylpyrrolidone (NMP) was first proposed by Coleman et al. [21], there have been several techniques of LPE reported, such as jet cavitation [22], high-shear mixing [19], microfluidization [23], etc. All of these techniques have their own advantages and limitations. Sonication assisted LPE has been widely used to prepare graphene, but suffers from high energy-extensive consumption and low efficiency [24]. Thus, it is not feasible for the scalable production of high-quality few-layer graphene. Luckily, it is shown that graphite can be exfoliated in the suitable solvents by shear force [25]. 

In this paper, we will review the latest progress on the preparation of graphene by liquid-phase exfoliation in organic solvents, ionic liquids, water/polymer or surfactant solutions, and some other green dispersants. Though several reviews [26,27,28,29,30,31] have already summarized the up-to-the-minute progress of the LPE, this review will focus on the latest high-shear and microfluidization assisted LPE in green dispersants. Moreover, several devices of high-shear mixing and exfoliation mechanisms will be also introduced in detail. Finally, we will give some perspectives on future research directions for the development of green exfoliation media and efficient techniques for producing high-quality graphene.

## 2. Techniques of Liquid-Phase Exfoliation

Traditionally, LPE includes mainly two different exfoliation techniques of graphite [24]: cavitation in sonication and shear forces in high-shear mixer. Recently, microfluidizer has been proven efficient to exfoliate graphite in suitable aqueous solutions under high shear rate [23]. LPE is a straightforward procedure with high potential for the mass production of graphene [19]. The basic devices of sonication or high-shear mixing are generally available. In addition, the operating conditions of LPE are mild and do not require vacuum or high temperature systems. However, the large-scale application of sonication assisted LPE has been hindered because of the low concentration of graphene and the high energy consumption during the production process. The high-shear mixing or microfluidizer method is an emerging LPE technology, which can exfoliate graphite successfully using high-shear mixer-driven fluid dynamics [14].

### 2.1. Sonication

Sonication is an effective exfoliation method and has the potential to produce monolayer or few-layer graphene at relatively high concentration. Typically, the power of sonication is used to induce physical or chemical changes in some systems through the generation of cavitation bubbles [32]. As the ultrasonic waves propagate through the medium, compressions and rarefactions exert high-pressure and low-pressure that pushes and pulls molecules. During rarefaction, microbubbles begin to form and grow with each cycle until they reach an unstable state and implode generating powerful shockwaves [33].

There are two types of sonication, namely bath sonication (BS) and tip sonication (TS). They have been employed simultaneously or individually to prepare monolayer or few-layer graphene sheets by exfoliating graphene [34]. Sonication-assisted LPE ordinarily involves three steps [35]: (1) the preparation of dispersion of graphite in a specific solvent; (2) the exfoliation of dispersion via sonication; (3) the purification of graphene. During sonication, the growth and collapse of the microbubbles in liquids is attributed to the cavitation-induced pressure pulsations. The effect of the cavitation results in high-speed microjets and shock waves, which will produce normal and shear forces on graphite [36]. The cavitation and shear forces play a significant role in the exfoliation of graphite to obtain graphene (Figure 5) [37]. The exfoliation effect depends on the power of sonication, the liquid medium used to disperse graphene nanosheets, and the rate of centrifugation [38]. The suitable liquid media are chosen to provide an environment that allows for stable dispersions of graphene during sonication. Different solvents have been used for LPE of graphite, and will be discussed in Section 3. Centrifugation is a process that removes large and unstably dispersed graphite particles or aggregates.

LPE of graphite was first carried out by Coleman et al. [21] via sonication in organic solvents, NMP. The dispersion of carbon nanotubes in organic solvents was successfully prepared by Bergin et al. [39], which is the driving force behind Coleman’s work. Coleman and his co-workers had been studied the effect of sonication time and centrifugation on the concentrations or lateral sizes of graphene nanosheets produced by sonication assisted LPE. The concentrations of graphene dispersions have been improved, up to 1.2 mg mL^−1^ and 4 wt % monolayers by using drastically longer sonication times (460 h) [40]. The graphene concentration after centrifugation is a function of sonication time (Figure 6), which closely follows C_G_∝t^1/2^. However, the dimension of the graphene flakes (∝t^−1/2^) is dramatically decreased with the increase of sonication time, but the intensity of the D band (I_D_) increases with time as t^1/2^, which means that sonication time can also have an effect upon the quality of graphene [35], but it indicates that new edges, rather than basal-planed defect, are formed. Moreover, Coleman et al. [41] also reported a method of separating the graphene dispersions having the average sheet length of about 1 μm into fractions, each with diverse average sizes. Centrifugation at high rates results in small flakes being dispersed but larger ones sedimenting out. Redispersion of sediment, followed by successive centrifugation, separation and redispersion cycles can be used to separate the flakes by size (Figure 7). This method is universal and can easily be applied to surfactant and polymer stabilized dispersion of graphene or any other layered materials [42] produced by high-shear mixing or microfluidization. 

The concentration of graphene dispersion produced via sonication in various solvents as shown in Table 2, the maximum concentration is 1.2 mg mL^−1^, which requires long time for sonication. So it is not viable for mass production of large amount of defect-free few-layer graphene via sonication. 

### 2.2. High-Shear Mixing

It has been shown that graphene can be exfoliated in the suitable liquid under shear force [25]. Moreover, the application of high shear forces through the use of high-shear mixers have been investigated as scalable methods of graphite exfoliation [19]. Shear exfoliation is similar to sonication exfoliation as aqueous liquids can be used to facilitate the exfoliation of graphite and provide stable dispersions of graphene thereby eliminating the use of harmful organic liquids. 

It is urgent to exploit industrially scalable methods for the productions of high-quality graphene by new exfoliation techniques expect sonication. In 2014, Coleman and co-workers made significant progress in the graphene production by shear exfoliation [19], which promoted the tremendous development of shear exfoliation technique. They demonstrated that high-shear mixing of graphite in appropriate solvents could prepare the high-concentrated dispersions of graphene nanosheets. The graphene flakes were unoxidized and without basal-plane defects. More importantly, when a local shear rate exceeded 10^4^ s^−1^, the exfoliation of graphite occurred in both laminar and turbulent regions. In particular, comparing shear exfoliation to sonication, shear exfoliation has higher efficiency (Figure 8). According to their scaling law, a rate of up to 100 g h^−1^ can be achieved when a liquid volume of 10 m^3^ is reached. Recently, the high-shear mixing was also applied to produce graphene oxide by the modified Hummers’ method [52].

Table 3 shows the concentration of graphene dispersion produced via high-shear mixing in various solvents, here the maximum concentration is 10 mg mL^−1^.

Until now, high-shear mixing exfoliation of graphite has been not completely studied, particularly, few studies focus on high-shear mixing in aqueous systems using ionic- or non-ionic surfactants. 

### 2.3. Microfluidization

Microfluidization is a high-pressure homogenization technique, where high pressure is exerted to the fluid and the pressure force drive the fluid to pass through the microchannel (diameter of microchannel, d < 100 μm) [60,61]. It provides mild exfoliation conditions that can help to decrease the formation of defects. Microfluidization is generally utilized in the nano emulsification [62], food industry [63], cell disintegration [64], the dispersion of carbon nanotubes [65], and active pharmaceutical ingredients [66]. Recently, microfluidizer was used to product graphene quantum dots (Figure 9) [67,68] and graphene-based conductive inks [23]. Compared to sonication and high-shear mixing, the main superiority of mircrofluidization is that high shear rate (>10^6^ s^−1^) exists in the whole fluid region.

Karagiannidis et al. [23] demonstrated the exfoliation of graphite in SDS aqueous solution by using microfluidic processor without additional washing or centrifugation (Figure 10). The yield of graphene nanosheets is 100% under high shear rate (10^8^ s^−1^) conditions. Conductive printable ink is formulated by using sodium salt of carboxymethyl cellulose (CMC-Na) stabilized graphene dispersion and the concentrations come up to 100 g L^−1^. In order to study effects of sonication time and microfluidization on the exfoliation efficiency, Wang et al. [60] demonstrated a novel and practical method to exfoliate natural graphite powder to obtain high-quality graphene by using a sequence of sonication and microfluidization in a mixture of NMP and NaOH. The maximum achieved few-layer graphene concentration is 0.47 mg mL^−1^ and the lateral sizes of the graphene flakes are in the extent of 0.5–2 μm.

As for the exfoliation mechanism during the microfluidization, there are three effects on the exfoliation of graphite—cavitation, shear stress, and collision (Figure 11) [60]. The dispersion of graphite was forced into the chamber channel, and the shear force controlled exfoliation. The collisions caused by micro jets could produce shear forces, in which monolayer or few-layer graphene could be self-peeled down owing to the lateral self-lubrication ability. The function of cavitation is similar to the sonication assisted LPE. 

Similarly, the technique of jet cavitation was proposed by Shen et al. [22,69] to exfoliate graphene in aqueous solution (Figure 12a). Within the nozzle (Figure 12b), Hydrodynamic-induced cavitation by high pressure difference and geometric-induced cavitation by abrupt geometric change enhance the intensity of jet cavitation [22]. The yield of graphene using this technique was ~4 wt % in water with surfactant and could be improved by changing the geometric shapes of nozzle. This device could be extended to prepare other 2D layered materials, such as BN [70] and MoS_2_ [71]. Some enormous improvements have been made [72] by means of the change of the treating time, jet pressure, and initial concentrations of graphite, the yield of graphene up to ~12 wt % and the maximum concentration up to 1.2 mg mL^−1^ were obtained in the aqueous solution of 75% acetone. However, micro-jet cavitation results in the absence of severe functionalization of graphene and the sizes of graphene sheets cannot be controlled. 

Recently, Shang et al. [73] presented the methods of the combining mic-jet cavitation and supercritical CO_2_. The yield of graphene nanosheets of less than three layers thickness reached 88%. The mechanism of exfoliation graphene by jet cavitation with or without supercritical CO_2_ is shown in Figure 13 [73] and Figure 14 [74]. Cavitation and shear force are the main exfoliation mechanism. Cavitation, just like the widely used sonication, occurs when fluid passes through the nozzle, which causes micro-jets and compression shock stress waves. The fluid in the nozzle forms turbulence because of the absence of high pressure difference at both ends of nozzle and the sudden geometric expansion and contraction of fluid channel, which leads to an inhomogeneous velocity distribution. The velocity gradient will result in the viscous shear force to exfoliate graphene. For micro-jet cavitation with supercritical CO_2_, there is one more mechanism, i.e., the intercalation and collision of supercritical CO_2_ molecules has a positive effect on the relaxation of Van Der Waals force among the graphite layers [75].

## 3. Media of Liquid-Phase Exfoliation

For LPE, one of the most important factors controlling the overall yield of exfoliation graphene is the appropriate solvents. The van der Waals interaction between the graphene layers held within a π–π stacking distance of 0.34 nm should be effectively overcome by the ideal solvent [76].

### 3.1. Organic Solvent

Previous studies have shown the interfacial tension plays a crucial role when a solid surface is immersed in a liquid medium [35]. The range of solvent surface tension that can better exfoliate graphene is 40–50 mJ m^−2^ [21] because they can minimize interfacial tension between graphene and solvent. Many original organic solvents have been used to exfoliate graphene (Table 4).

In addition to the above common organic solvents, a series of perfluorinated aromatic molecules [80], including C_6_F_6_, C_6_F_5_CF_3_, C_6_F_5_CN and C_5_F_5_N, has been reported. The exfoliation capability of each solvent in ascending order was as follows: C_6_F_5_CF_3_≈C_5_F_5_N<C_6_F_6_<C_6_F_5_CN. Hence, C_6_F_5_CN provided the highest yield and concentration of no oxidization of graphene dispersion (2%, 0.1 mg mL^−1^). Moreover, Qian et al. [81] reported the monolayer or few-layer graphene dispersions could be obtained by exfoliating the expanded graphite in acetonitrile (ACN), whose yield was 10 wt %–12 wt %. Organic amine-based solvents (DMPA, DMAPMA, BAEMA, MAEMA), which perform previously used solvents such as NMP in terms of their dispersing capacity, were discovered which enable direct exfoliation of graphite to produce high-quality and oxygen-free graphene nanosheets [82].

Although the LPE has been rapidly developed in the past decade, many severe problems still exist. One of the most serious challenge is the lover concentration of graphene dispersions obtained in the pure organic solvents, generally, less than 0.01 mg mL^−1^ [46]. Besides, the ideal organic solvents discussed above are not volatile and have high boiling points [51], which is exceedingly difficult to remove residual organic solvents, causing tricky problems for subsequent processing [21], such as flake deposition and composite formation [51].

Fortunately, the efficiency of exfoliation can be improved by adding auxiliary agents, such as inorganic salt [46] and organic salt [83], or just increasing the time for sonication or high-shear mixing. A simple method for producing graphene by bath sonication assisted liquid-phase exfoliation of natural graphite in organic solvents with NaOH was reported by Liu et al. [46]. It is found that the addition of NaOH can apparently improve the exfoliation efficiency of graphene, up to 20 times in the cyclohexanone (CYC) (Figure 15), owing to the intercalation of Na^+^ and OH^-^ increasing the interlayer spacing of graphite. Du et al. [83] demonstrated an efficient method to exfoliate pristine graphite in common organic solvents (NMP, DMF, and DMSO) with the addition of organic salt (Na_3_C_6_H_5_O_7_, Na_2_C_10_H_14_N_2_O_8_, Na_2_C_4_H_4_O_6_, and KNaC_4_H_4_O_6_) (Figure 16) These organic salts can significantly improve the efficiency of exfoliation graphene and the concentration of graphene dispersions in DMSO with Na_3_C_6_H_5_O_7_ obtained just via a short 2 h sonication was 0.72 mg mL^−1^. Thus, DMSO can be used as a suitable exfoliation medium instead of NMP and DMF. More importantly, the monolayer or few-layer graphene nanosheets obtained were without defects and oxides. Naphthalene [84], serving as a ‘molecular wedge’ to intercalate into the edge of graphite to promote the exfoliation of graphite, also has been studied as an additive to prepare graphene dispersions in common organic solvents. The results have shown that naphthalene could increase the yield of graphene (Figure 17), among which the concentration of graphene dispersion was doubled in NMP with naphthalene, up to 0.15 mg mL^−1^ after 90 min sonication. 

Some researchers try to exfoliate graphene in low boiling points solvent in order to overcome the limitations of using high boiling solvents. Zhang et al. [85] demonstrated a solvent exchange method, the concentration of graphene dispersion in ethanol up to 0.04 mg mL^−1^. Coleman et al. [51] demonstrated the exfoliation of graphite in low boiling point solvents, including acetone, isopropanol (IPA), and chloroform. The final concentrations of graphene, without defects in the basal plane, was 0.5 mg mL^−1^ by optimizing the parameters of the dispersion procedures (Figure 18). However, the concentrations of graphene dispersions were lower than that of in common organic solvent and some aggregation was observed. Choi et al. [86] reported a preparation method of graphene dispersions (0.025 mg mL^−1^) in 1-propanol (23.4 mJ m^−2^), a volatile solvent, which could be removed just by air drying without heating. This means that the surface tension is not always a valid criterion for predicting the dispersibility of graphene in solvents, which should be characterized by the Hildebrand or Hansen solubility parameters [87].

### 3.2. Ionic Liquids 

Ionic liquids, an organic molten salt, having a melting point below 100 °C [88,89], usually have high property to dissolve various solutes. Interestingly, ionic liquids have the potential to stabilize the exfoliated graphene because of their iconicity. 

Wang et al. [90] demonstrated that non-oxidized high-concentration (up to 0.95 mg mL^−1^) graphene dispersion can be produced via sonication in 1-butyl-3-methyl-imidazolium bis (trifluoro-methane-sulfonyl) imide, an ionic liquid. Ionic liquids that containing non-aromatic cations gave similar results. Nuvoli et al. [91] found 1-hexyl-3-methylimidazolium hexafluorophosphate, a commercially available ionic liquid, could be also used to exfoliate graphene via sonication. The concentration of graphene dispersion obtained is up to 5.33 mg mL^−1^, without any chemical modification. Ager et al. [92] demonstrated essentially complete exfoliation of graphene aggregates in water at concentrations up to 5% by weight using recently developed triblock copolymers and copolymeric nanolatexes based on a reactive ionic liquid acrylate surfactant. 

High concentration of graphene dispersion can be obtained in ionic liquids, but ionic liquids are expensive and difficult to remove, which limits its wide application. In addition, the viscosity of the ionic liquids is large, which will have a certain effect on the exfoliation of graphene.

### 3.3. Water/Surfactant

The exfoliation of graphene in organic solvents or ionic liquids is facile, involving only two phase dispersions [26,27], namely graphene and solvents. However, the amount of available solvents that are propitious to liquid-phase exfoliation of graphene is limited. In addition, the solvents that can be used to disperse graphite are relatively expensive and toxic. Water is usually considered to be the ideal dispersion medium [93], and the natural hydrophobicity and limitations to scalable production of 2D nanoplatelets have become a considerable obstacle to their application in various fields. The high surface tension of pure water, 72.8 mJ m^−2^, can be reduced by the addition of suitable surfactants so that water/surfactant system has potential to apply to the exfoliation and stabilization of graphene (this review focuses only on water/surfactant system considering the limitations of organic solvents). Actually, surfactants in aqueous solutions are more environmentally friendly while producing comparable concentrations of graphene to pure solvent exfoliation. 

The exfoliation of graphite in surfactant solutions is one of the most promising methods to produce high-quality graphene [94]. A range of studies have shown surfactant-assisted exfoliation is a promising method for preparing high concentration graphene dispersions. Lotya et al. [95] firstly reported a method to obtain exfoliated non-oxidative graphene via 30 min sonication in water/surfactant (sodium dodecylbenzene sulfonate, SDBS) solutions, the concentration of graphene dispersion was up to 0.002–0.05 mg mL^−1^, which was similar to that of nanotube dispersion in the surfactant aqueous solution [39,96]. Green et al. [97] shown that the surfactant sodium cholate (SC), a conventional anionic surfactant, can be applied to prepare stable graphene dispersions, the concentration of graphene exceeded 90 µg mL^−1^. Coleman et al. [98] dispersed graphene in a range of ionic or non-ionic surfactants aqueous solution, the concentration of dispersion obtained is 11–26 µg mL^−1^.

In order to improve the concentration of graphene dispersion, long-time sonication (up to 400 h) was utilized in SC aqueous solution, the final concentration is up to 0.3 mg/mL [34]. Multi-step sonication procedure (tip sonication–bath sonication–tip sonication) significantly improves the concentration of exfoliated few-layers graphene nanosheets in various dispersants (Figure 19) [50]. The maximum concentration graphene nanosheets received was 0.7 mg mL^−1^ by increasing the total sonication energy/volume and bath sonication energy/volume (Figure 20). Sun et al. [99] reported a vital method to afford high-concentration graphene (up to 7.1 mg mL^−1^) in the aqueous solution of sodium taurodeoxycholate (STC), an anionic surfactant, via only 24 h tip sonication. A novel anionic surfactant (sulfonated used engine oil (SUEO)), which was prepared from used engine oil, was employed to exfoliate graphene sheets under sonication. The exfoliating and dispersing efficiencies of SUEO, estimated by the concentration of graphene dispersions (0.477 mg mL^−1^), were higher than that of traditional surfactants, such as SDS, SDBS, CTAB, and PVP.

As a matter of fact, any surfactant promotes exfoliation (Figure 21) [100], among which, Pluronic^®^ P123 provided the highest concentration of graphene dispersions (up to 1 mg mL^−1^). LPE in non-ionic surfactant systems, providing better stability and higher concentrations of graphene nanosheets, performed better than that of in ionic surfactants. However, surfactants were added only once at the beginning of sonication or high-shear mixing, the concentration of exfoliated graphene was too low to be used for a potential industrially scalable process. Fortunately, Notley et al. [101] successfully prepared high-concentration graphene dispersions (10.23 mg mL^−1^) by the continuous addition of Pluronic^®^108, a nonionic surfactant, this method could be also used with other surfactants (Table 5).

In summary, surfactant-assisted LPE of graphite can offer high-concentration defect-free graphene. However, the disadvantage of using surfactants over organic solvents is because of the adsorption of surfactants onto the surface of the graphene. While the adsorption process could prevent the re-aggregation of graphene sheets through hydrophilic interactions [102], an extra processing step is required to remove adhered surfactants before the graphene is used as intended. 

### 3.4. Water/Polymers

The application of non-ionic and non-toxic polymers as assistant in the preparation of graphene has also made progress. For the exfoliation procedure, the exfoliation of graphene in the water/polymer is similar to that of in the water/surfactants. The stabilization mechanism of exfoliated nanosheets is the main difference between the two methods. The stability of the most graphene dispersions exfoliated in polymer aqueous solution is provided by the combination of steric factors and non-covalent interactions. 

Bourlinos et al. [103] first reported that polyvinylpyrrolidone (PVP), a non-ionic, inexpensive, and non-toxic polymer, could be utilized to exfoliate graphite offering few-layer graphene (0.1 mg mL^−1^) without oxidation in the aqueous phase via sonication. Sodic carboxymethylcellulose (CMC-Na) and DNA showed the similar behavior. The dispersions of PVP-stabilized graphene obtained by tip sonication was stable and expediently re-dispersible in series of organic solvents (VP, NMP, DMF, DMSO, ethanol, methanol) [48], among which the maximum concentration of graphene dispersions was up to 0.7 mg mL^−1^ in vinyl pyrrolidone (VP). Sun et al. [104] demonstrated that few-layer high-concentration graphene (up to 4 mg mL^−1^) can be prepared in low boiling point alcohols, ethanol and isopropanol, with the addition of polymer P20 via bath sonication. Phiri et al. [24] found that concentration of few-layer graphene dispersions was up to 0.7 mg mL^−1^ by 120 min high-shear mixing in PVP aqueous solution, which means that the efficiency of high-shear mixing is several times than that of sonication exfoliation [102]. Modified acrylate polymers can also effectively exfoliate and stabilize pristine graphene nanosheets in aqueous media [105].

Besides, it has been recently shown that graphite can be exfoliated in pure water or organic solvent by adding a various of conventional polymers, such as polystyrene(PS) [106], ethyl cellulose (EC) [107], polystyrene-co-butadiene (PBS), polymethyl methacrylate (PMMA) [108], cellulose acetate (CA) [109], polyetherimide (PEId) [110], polypropylene(PP) [111], hyperbranched polyethylene (HBPE) [112], polylactide (PLA) [113], and so on.

The use of polymers in the LPE process is no doubt better than that of pure organic solvents. However, the majority of graphene prepared by polymer-assisted LPE cannot be extracted from graphene–polymer composites on account of the strong interactions between graphene and polymer. 

### 3.5. Direct Exfoliation in Pure Water 

In general, polymers or surfactants as stabilizers are challenging to be removed, which results in the poor performance of the graphene prepared by polymer- or surfactant-assisted LPE. The extensive applications of graphene are limited by the lack of cost-effective massive production methods [114]. Therefore, it is vital and urgent to research how to directly disperse graphene in pure water, a common green solvent, without any additional stabilizers. 

Shen et al. [93] reported that water could steadily disperse graphene prepared by LPE. The liquid-exfoliated graphene can be well dispersed in pure water to form steady dispersions of certain concentration just via mild bath sonication, which owing to the reduction of the size of nanosheets and the resulting enhancement of edge effects. However, this method cannot be applied to exfoliate graphite to prepare graphene. Kim et al. [115] found bulk-layered materials—such as graphene, MoS_2_, MoSe_2_ [116], and h-BN—could be exfoliated and dispersed by just dominating the temperature of bath sonication and storage in pure water without any the addition of chemicals or surfactants. The higher temperatures were propitious to the exfoliation and stability of graphene dispersions, and the final concentration arrived up to 0.0065 mg mL^−1^ (Figure 22). The graphene was functionalized on its edges at the high temperature sonication so that functional groups improved the solubility of graphene in pure water. Molecular dynamics (MD) simulations were carried out to study the solvation states of graphene (including –OH or –COOH functionalized graphene), h-BN and MoS_2_ (Figure 23). The charge distribution and charge polarity on the surfaces of those materials can increase the interactions with water molecules. Lin et al. [37] demonstrated the starting natural graphite could be exfoliated to product well-dispersed few-layers graphene in pure water, which was free of significant defects, by ozone-assisted sonication without any the addition of chemical reagents. More importantly, this method is promising for preparing various bulk-layered materials. The graphite pretreated by vapor can be used to product the edge hydroxylated graphene nanosheets in pure water without the addition of any surfactants or polymers by sonication-assisted liquid-phase exfoliation (Figure 24) [117]. The concentration of stable graphene dispersion arrived up to ~0.55 mg mL^−1^.

### 3.6. New Type of Green Dispersants

Recently, some novel green dispersants—such as ammonia solution [49], alkali lignin [118], black liquor [53], black tea [54], etc.—have been proven to be able to achieve the exfoliation of natural graphite to prepare few-layer defect-free graphene nanosheets.

Shen et al. [49] reported graphite could be exfoliated to prepare few-layer high-quality graphene in water (the concentration up to 0.058 mg mL^−1^) with the addition of ammonia solution, which released the gaseous ammonia playing a pivotal role in the graphene exfoliation. This preparation method was simple and did not involve any organic solvents, surfactants or polymers. In order to optimize the exfoliation strategy, the starting graphite concentration and sonication time (t_s_) were investigated, C_G_ = 20.28 t_s_^1/2^ (C_G_, the concentration of graphene), the results accorded with that presented by Coleman’s research group (C_G_∝t_s_^1/2^) [40]. In general, the higher the rate of centrifugation was, the smaller the lateral size of graphene was. The graphene concentration was 113.69 μg mL^−1^ at 1000 rpm, while it reduced to 13.80 μg mL^−1^ at 3000 rpm.

Lignin, a low-cost by-product of the pulp and paper industry, has been utilized directly or with chemically modified as dispersant agent [119]. Recent research shows lignin peroxidase [120] could be applied to degrade the oxidized and reduced graphene nanoribbons. More interestingly, the liquid-phase exfoliation of graphite can be carried out in aqueous media with the addition of alkali lignin via bath sonication [118]. The considerable concentration of single- or few-layer graphene reached up to 0.65 ± 0.03 mg mL^−1^. Similarly, pulping black liquor, a main source of papermaking pollution, contains about 65% alkali lignin. The treatment technology of black liquor is complex, expensive, and may bring secondary pollution. Fortunately, Ding et al. [53] found layered materials, such as graphene, MoS_2_ and BN, could be efficiently exfoliated by high-shear mixing in pulping liquor. This green, environmentally friendly method not only can solve the pollution of water liquor, but also give high-concentration dispersions of 2D layered materials (6.0 mg mL^−1^ for BN, 6.3 mg mL^−1^ for MoS_2_ and 10 mg mL^−1^ for graphene) (Figure 25). These single- or few-layered nanosheets can be dispersed well because of the functionalization of edges. The yields of BN, MoS_2_, and graphene obtained by shear-assisted exfoliation approximately linearly increases with the exfoliation time. These results indicate that pulping black liquor can be used as a favorable dispersant for the mass production of layered materials. Tea has been studied as a green reduction agent of graphene oxide to graphene (Figure 26) [121,122]. Ismail et al. [54] found that the commercial black tea could be applied to directly exfoliate graphene in a kitchen mixer. The maximum concentration of graphene was up to 0.032 mg mL^−1^ via only 15min high-shear mixing, which could be increased by changing the concentration of starting graphite or the time of high-shear mixing. The value of I_D_/I_G_ from Raman date is 0.17, which is very minor compared with that of reduced graphene oxide (generally, I_D_/I_G_ > 1), despite the probable functionalization of graphene by oxygen groups in black tea.

In addition to the direct exfoliation of flake graphite above, another prevalent method of graphite exfoliation is the conversion of graphite into graphite intercalation compounds (GICs). The intercalation can extend the graphite so as to reduce the adhesive forces among the graphene layers. Therefore, GICs can be exfoliated easily by sonication or high-shear mixing in the appropriate solvents [123]. For example, Mahmoud et al. [124] recently presented a new efficient method for the exfoliation of graphite to obtain defect-free graphene under high temperature (2000 °C). The preparation process is shown in Figure 27. Firstly, the FeCl_3_-graphite intercalation compounds were treated with dodecylamine (DA) at 90 °C to product amine-treated graphene (G-90), and DA could be completely removed through the subsequent heating treatment; secondly, G-90 was heated at 900 °C in air to product graphene (G-900); Finally, G-900 was further heated at 2000 °C under nitrogen atmosphere to obtain well-expanded graphene (G-2000). Though the exfoliation efficiency of GICs is higher than the direct exfoliation of flake graphite, this requires a special intercalation process.

## 4. Devices of Liquid-Phase Exfoliation

This review concentrates on the devices that can offer high-shear force to mildly exfoliate graphene, such as the Taylor–Couette flow reactor, rotor-stator mixer, rotating-blade mixer, and microfluidizer [61]. Meanwhile, as these equipment have their own superiority and drawbacks, it is still urgent to design an appropriate equipment for liquid-phase exfoliation of graphene. 

### 4.1. Taylor-Couette Flow Reactor

Taylor-Couette flow reactor [55], a vortex fluidic device, can generate Taylor vortex flow [125] in the narrow space between two concentric cylinders (Figure 28) to exfoliate natural graphite flakes by the shearing force of less energy intensity in an organic solvent, NMP, this process does not involve sonication-induced cavitation. The concentration of graphene reached 0.15 mg mL^−1^ after 100min of mixing time (the highest concentration near 0.65 mg mL^−1^ when the initial graphite concentration is 50 mg mL^−1^) and the yield of high-quality (the value of I_D_/I_G_ = 0.14) few-layer (90% of flakes less than five layers) graphene can achieve up to 5% by optimizing the parameters of vortex flow process based on computational fluid dynamics.

Taylor vortex flow can generate the high local shear rate and high wall shear stress in each vortex cell at the same time, which is potentially responsible for the efficient exfoliation of the starting graphite flakes [25].

### 4.2. Rotor-Stator Mixer

Paton et al. [19] first reported the starting graphite can be exfoliated to prepare high-quality (the value of I_D_/I_G_ = 0.18) graphene by Silverson model L5M mixer (Figure 29), a rotor-stator mixer, which generated high shear in the narrow gap (~100 μm) between the rotor and stator, in organic solvent (NMP) and surfactant (sodium cholate) aqueous solutions. The production rate of graphene (P_R_ = VC/t, the maximum value was 1.44 g h^−1^ using this laboratorial mixer) using the method scales as P_R_∝V^1.6^ for the surfactant and P_R_∝V^1.1^ for NMP, which means the scalable production can be achieved just by increasing the volume of liquid and surfactants were more promising for scalable production.

Shen et al. [56] reported graphite could be exfoliated to product defect-free graphene nanosheets in 40 vol % IPA-water mixtures using FM300, a high shear mixer, made by Fluko in China, the concentration of single- or few-layer graphene reached 0.27 mg mL^−1^. The shear exfoliation of graphite was carried out in PVP and SC aqueous solution using an available high-shear colloidal mixer, which was equipped with high-shear generator system of three-stage rotor-stator (Figure 30) to optimize the capability of exfoliation [24]. There are also some other types of rotor-stator mixers which are used to provide high-shear force to induce the exfoliation of graphite, such as FJ300-S model high-shear mixer [53], FA40 model high-shear mixer (Fluko) [126], etc.

During mixing, the high-speed rotation of the rotor blades in the workhead serves as a pump exerting a powerful suction to draw in the liquid and graphite (Figure 31). Then, centrifugal forces drive graphite to the periphery between rotor and stator. High-shear forces are exerted to graphites when they high-velocity pass through the perforations in the stator and back into the main body of mixer. Graphite can be successfully exfoliated by the high-shear forces because of the continual rotation of rotor [19,53]. In the exfoliation process, not only high-shear forces play a major role, but also the collision and micro-jet cavitation can have an effect on the graphite exfoliation (Figure 32) [56].

### 4.3. Rotating-Blade Mixer

Though the exfoliation of graphene using rotor-stator mixer can achieve scale-up by just increasing the volume of liquid, the graphene concentration, C∝t*^n^* (*n* < 1), means that production rate declines with mixing time. This problem may be bound up with the details of the interaction of liquid and rotor-stator [57].

To explore a new equipment to offer high shear, Shen et al. [127] first reported the exfoliation of graphite to prepare high-quality (the value of I_D_/I_G_ < 0.12, free of basal-plane defects or oxidation) graphene by using a kitchen blender, a rotating-blade mixer, the concentration reached up to 0.22 mg mL^−1^ and the yield was 7.3% after 8 h of high-speed mixing. Similarly, Coleman et al. [57] utilized kitchen blade mixer and household detergent to prepare graphene, and the concentration was unexpectedly up to 1 mg/mL after a few hours mixing in such simple exfoliation system. Pattammattel et al. [128] reported the application of kitchen and edible proteins in pure water to prepare high-concentration low-defect graphene dispersions without oxidation. For bovine serum albumin (BSA) assisted exfoliation, inconceivably, gave the maximum concentration up to 6.8 mg mL^−1^, which was much higher than those generally reached via shear or sonication exfoliation, owing to the strongest negative charge of BSA in all the proteins used. For the shear exfoliation using the rotating-blade mixer, the concentration of graphene dispersion increases linearly with time so that it is 10 times higher than those reached using rotor-stator mixer. More importantly, the concentration declines tardily with volume, which results in the production rate increasing with volume [57].

For exfoliation mechanism of rotating-blade mixer, it involves multifarious fluid dynamics events, such as turbulence (turbulence is mainly in charge of high-shear rates which are not localized in any single part of the vessel [57]), shear, and collisions [127]. These fluid dynamical events or their corresponding effects primarily generate lateral-forces which preside over the exfoliation mechanism (Figure 33). As for microfluidizer, a high-pressure homogenizer, offering high shear, has been introduced in detail in Section 2.3.

## 5. Conclusions and Perspectives 

It is now well-acknowledged that one of critical bottlenecks on some important classes of applications—such as printed electronics, conductive coatings, and composite fillers—is the lack of industrial-scale methods to produce high-quality graphene in the form of liquid suspensions, inks, or dispersions [10]. LPE is just a promising cost-effective method for scalable production of single- or few-layer graphene through simple and available devices [129]. Since 2008, when LPE of graphite powder via the sonication was first initiated [21], huge progress has been made in the past decade. Sonication-assisted LPE has been widely used to prepare the dispersions of graphene, but suffers from high energy-extensive consumption and low efficiency. Fortunately, the appearance of high-shear mixing and microfluidization recently brings new vitality to the development of LPE.

This review has highlighted the latest progress on the successful preparation of graphene in various media, including organic solvents, ionic liquids, water/polymer or surfactant solutions, and some other green dispersants. The techniques of LPE, namely, sonication, high-shear mixing and microfluidization have also been reviewed subsequently. Moreover, several typical devices of high-shear mixing and exfoliation mechanisms were also introduced in detail. This systematically exploratory study of LPE will pave the way for the scalable production of graphene, which can be applied to produce other 2D layered materials, such as BN, MoS_2_, WS_2_, etc. However, it is still an enormous challenge to achieve fundamental goals of commercial availability of massive high-quality graphene with cost-efficient and environmentally friendly LPE. There are multitudinous conundrums to be further studied and overcome, some of which and our perspectives are discussed as follows:

The yield of LPE is still too low to meet the macroscopical requirement of industrial application of graphene, and the production of single-layer, large-size graphene is a huge challenge as well. The amount of available solvents that are propitious to LPE of graphene is limited. In addition, most solvents that can be used to disperse graphite are relatively expensive and toxic. Polymers or surfactants as dispersants are challenging to be removed, which results in the poor performance of the graphene prepared by polymer- or surfactant-assisted LPE. Sonication or shear exfoliation generally result in the reduction of the size of graphene nanosheets. The devices of high shear mixing are utilized directly from other fields of science, there is no equipment wholly designed for LPE. Graphene nanosheets prepared by LPE tend to aggregate easily and irreversibly in most solvents due to van der Waals forces and high surface energy, so separating graphene from the dispersion is another important issue [130]. More importantly, there is a more basic and systematic understanding of the advanced LPE mechanisms, which would lead to more innovative ideas about new and more efficient methods.

Though the specified centrifugation strategy [41] can separate the graphene dispersions with different average sheet size, the method cannot solve the fundamental problem, i.e., the uncontrollable size of graphene nanosheets obtained by LPE. The exploration of new green cost-efficient exfoliation media and techniques to increase the exfoliation efficiency of graphite is urgent and vital; The pretreatment of the starting materials may be very good way, such as the conversion of graphite into graphite intercalation compounds [131], or direct exfoliation in pure water by controlling the conditions of exfoliation, such as temperature and pressure, which can not only prevent the agglomeration of the nanosheets, but also simplify the exfoliation process or increase the yield of graphene. The stirred-tank reactor, a common equipment in chemical industry, may be effective when applied to exfoliate graphene. The combination of high-shear mixing with chemical methods or supercritical fluid techniques may further improve the efficiency of exfoliation or the quality of graphene. The method of molecular dynamic simulation can be used to further explain the advanced mechanisms for optimizing LPE methods [22].

## Figures and Tables

**Figure 1 nanomaterials-08-00942-f001:**
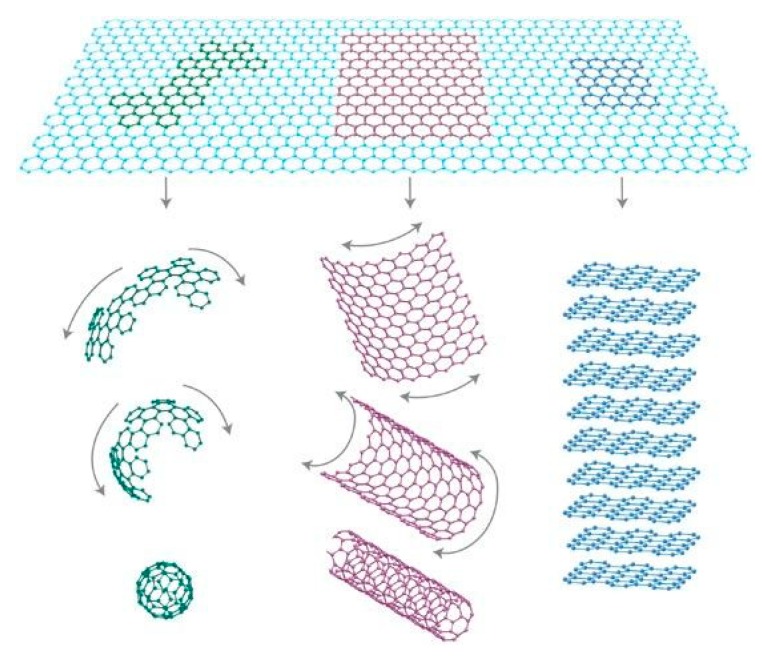
Mother of all graphitic forms. Graphene is a 2D building material for carbon materials of all other dimensionalities. It can be wrapped up into 0D buckyballs, rolled into 1D nanotubes or stacked into 3D graphite. Reproduced from [1]. Copyright Nature Publishing Group, 2007.

**Figure 2 nanomaterials-08-00942-f002:**
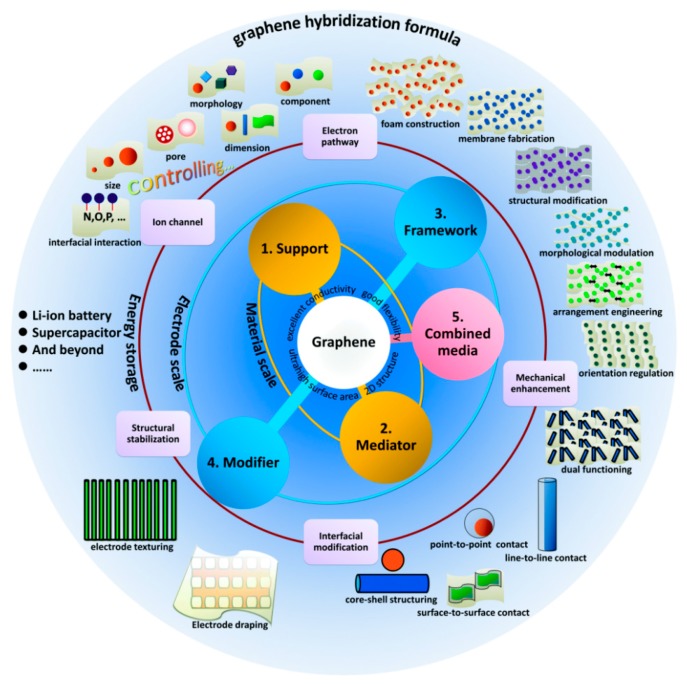
Overview of graphene hybridization formulas for energy storage applications. Reproduced from [7]. Copyright Royal Society of Chemistry, 2018.

**Figure 3 nanomaterials-08-00942-f003:**
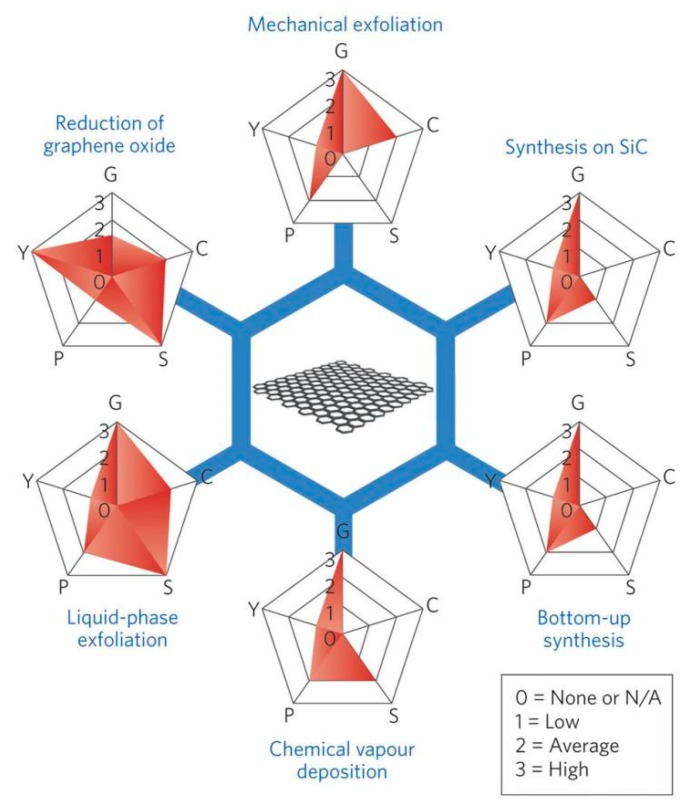
Schematic of the most common graphene production methods. Each method has been evaluated in terms of graphene quality (G), cost aspect (C; a low value corresponds to high cost of production), scalability (S), purity (P), and yield (Y) of the overall production process. Reproduced from [12]. Copyright Nature Publishing Group, 2015.

**Figure 4 nanomaterials-08-00942-f004:**
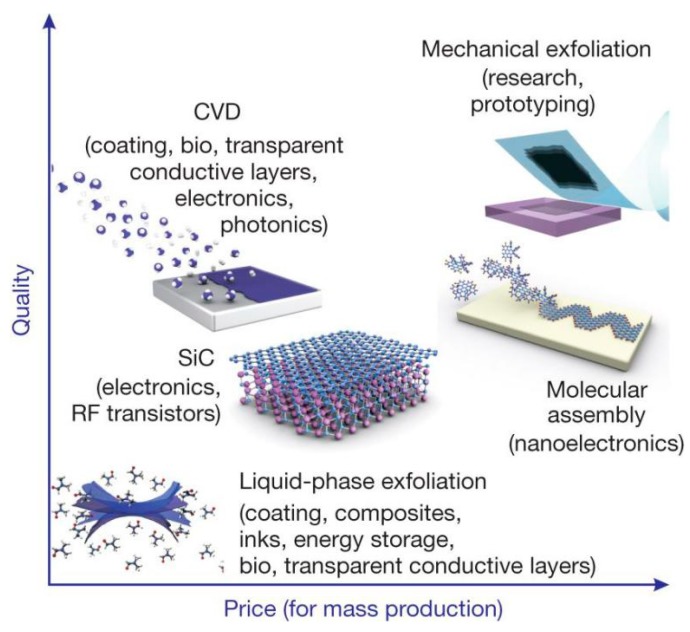
There are several methods of mass-production of graphene, which allow a wide choice in terms of size, quality and price for any particular application. Reproduced from [9]. Copyright Nature Publishing Group, 2012.

**Figure 5 nanomaterials-08-00942-f005:**
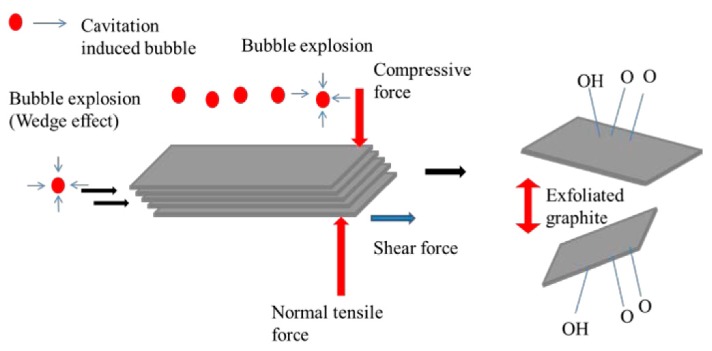
Illustration of the possible mechanism of graphite exfoliation. Reproduced from [37]. Copyright Multidisciplinary Digital Publishing Institute, 2016.

**Figure 6 nanomaterials-08-00942-f006:**
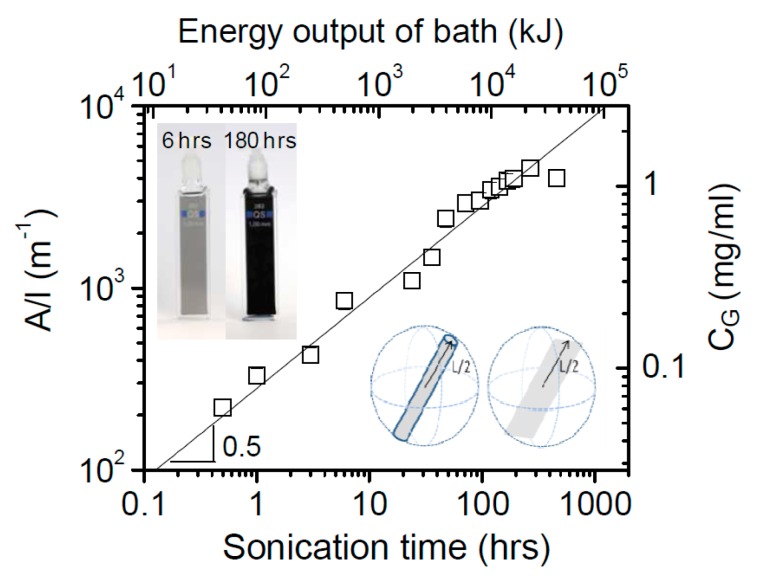
Concentration of graphene in NMP after centrifugation as a function of sonication time. Reproduced from [40]. Copyright Wiley-VCH, 2010.

**Figure 7 nanomaterials-08-00942-f007:**
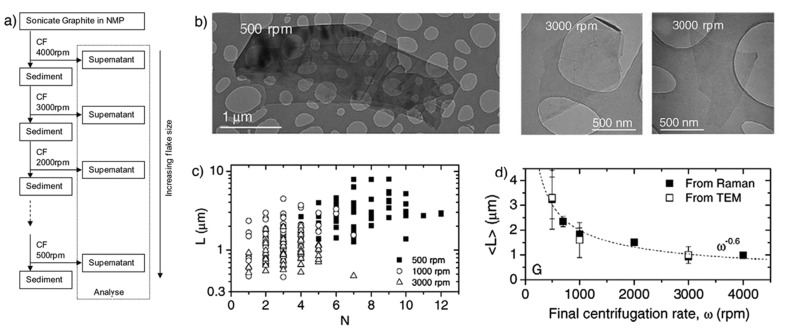
(**a**) Schematic representation of the flake separation process; (**b**) TEM images of flakes prepared at final centrifugation rates of 5000 rpm and 3000 rpm; (**c**) individual flake length plotted versus estimated flake thickness (number of monolayers, N) for dispersions with final centrifugation rates of 500, 1000, and 3000 rpm; (**d**) mean flake size as measured from TEM images. Reproduced from [41]. Copyright Elsevier, 2012.

**Figure 8 nanomaterials-08-00942-f008:**
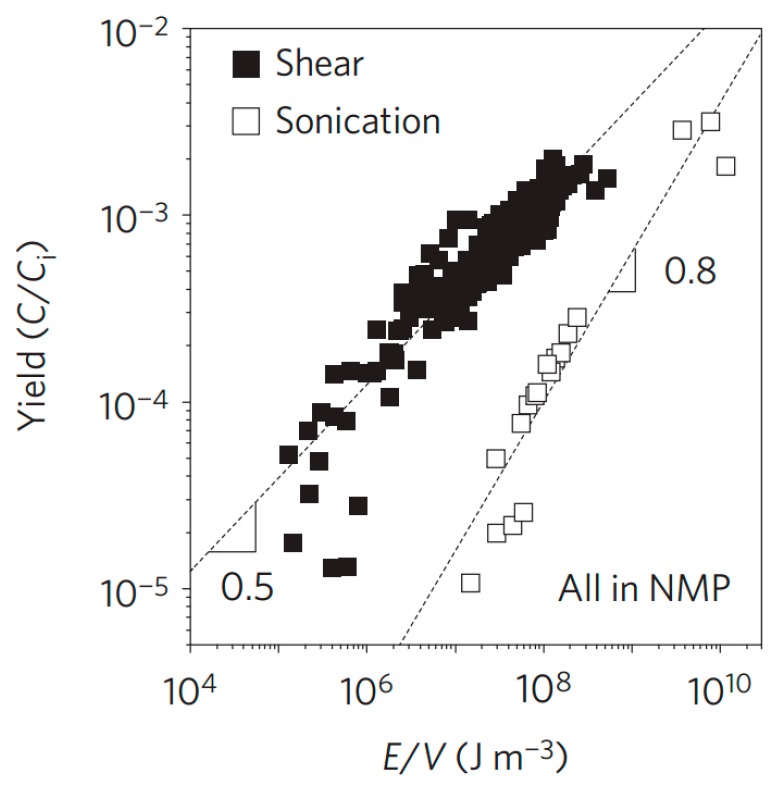
Graphene yield measured versus energy density. Reproduced from [19]. Copyright Nature Publishing Group, 2014.

**Figure 9 nanomaterials-08-00942-f009:**
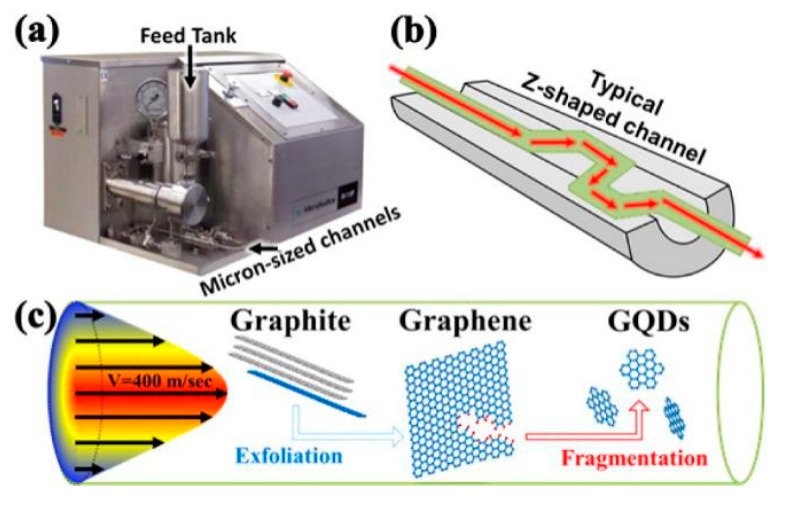
(**a**) Photograph of typical microfluidizer. The graphite aqueous suspension is powered by 30 kpsi pump from the feed tank through microsized channels. Schematics of (**b**) Z-shaped channels with diameters ranging from 400 μm down to 87 μm, and (**c**) typical flow profile within the channel with maximal flow speed of 400 m s^−1^. The graphite flakes are exfoliated into graphene sheets and further fragmented into nanosized GQDs. Reproduced from [67]. Copyright American Chemical Society, 2016.

**Figure 10 nanomaterials-08-00942-f010:**
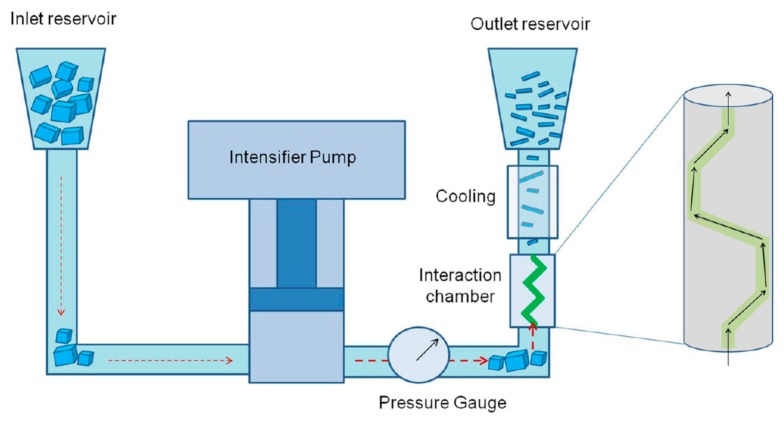
Schematic of the microfluidization process. An intensifier pump applies high pressure (up to ~207 MPa) and forces the suspension to pass through the microchannel of the interaction chamber where intense γ 10^8^ s^−1^ is generated. The processed material is cooled and collected from the outlet reservoir. The process can be repeated several times. Reproduced from [23]. Copyright American Chemical Society, 2017.

**Figure 11 nanomaterials-08-00942-f011:**
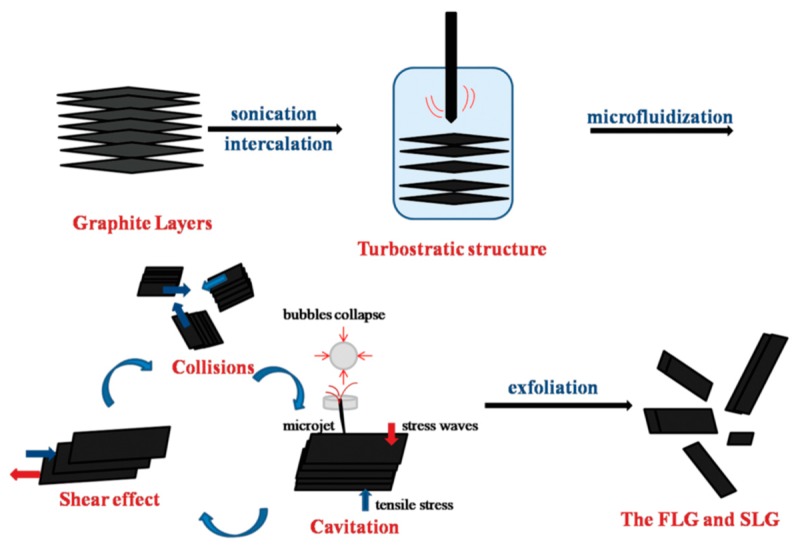
Exfoliation process and mechanism. Reproduced from [60]. Copyright American Scientific Publishers, 2017.

**Figure 12 nanomaterials-08-00942-f012:**
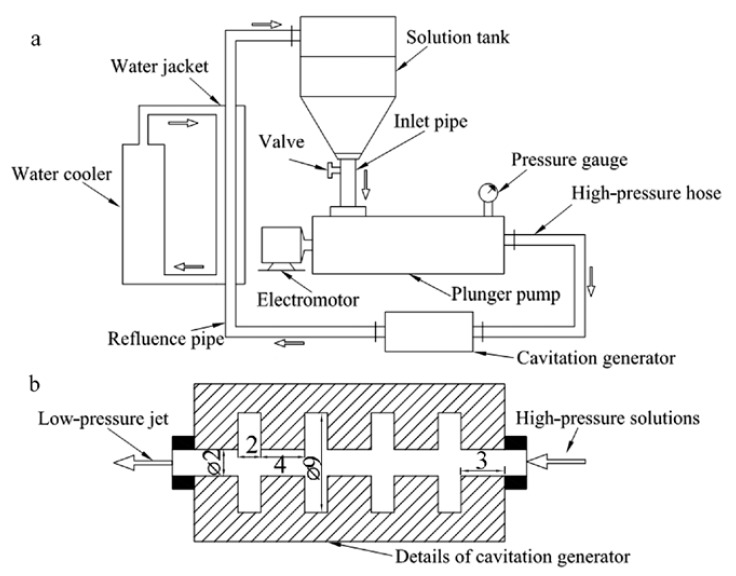
(**a**) Schematic of the designed jet cavitation device. (**b**) Details of the cavitation generator (mark unit: mm). Reproduced from [22]. Copyright IOP Publishing, 2011.

**Figure 13 nanomaterials-08-00942-f013:**
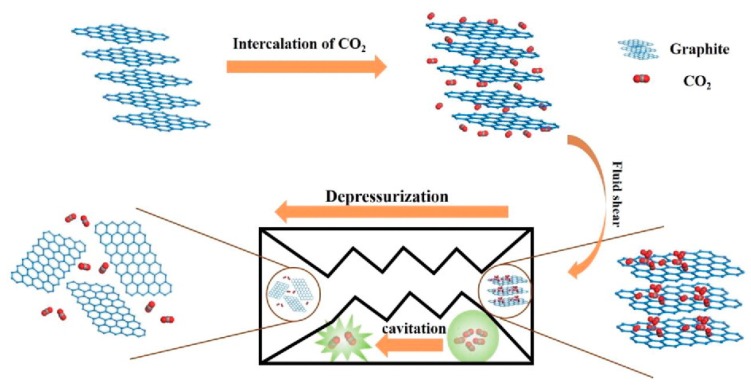
The mechanism drawing of exfoliation graphene nanosheets with supercritical CO_2_ and micro-jet exfoliation. Reproduced from [73]. Copyright IOP Publishing, 2017.

**Figure 14 nanomaterials-08-00942-f014:**
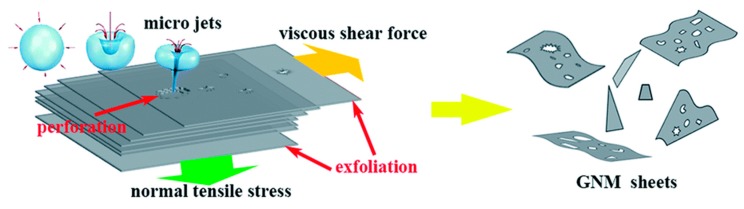
A schematic illustration showing the exfoliation mechanism of the jet cavitation. Reproduced from [74]. Copyright Royal Society of Chemistry, 2014.

**Figure 15 nanomaterials-08-00942-f015:**
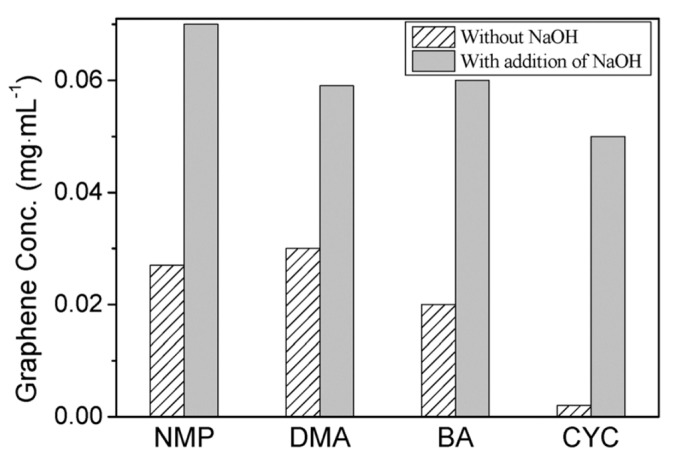
Concentrations of graphene sheets dispersed in organic solvents with or without addition of NaOH. Reproduced from [46]. Copyright Royal Society of Chemistry, 2011.

**Figure 16 nanomaterials-08-00942-f016:**
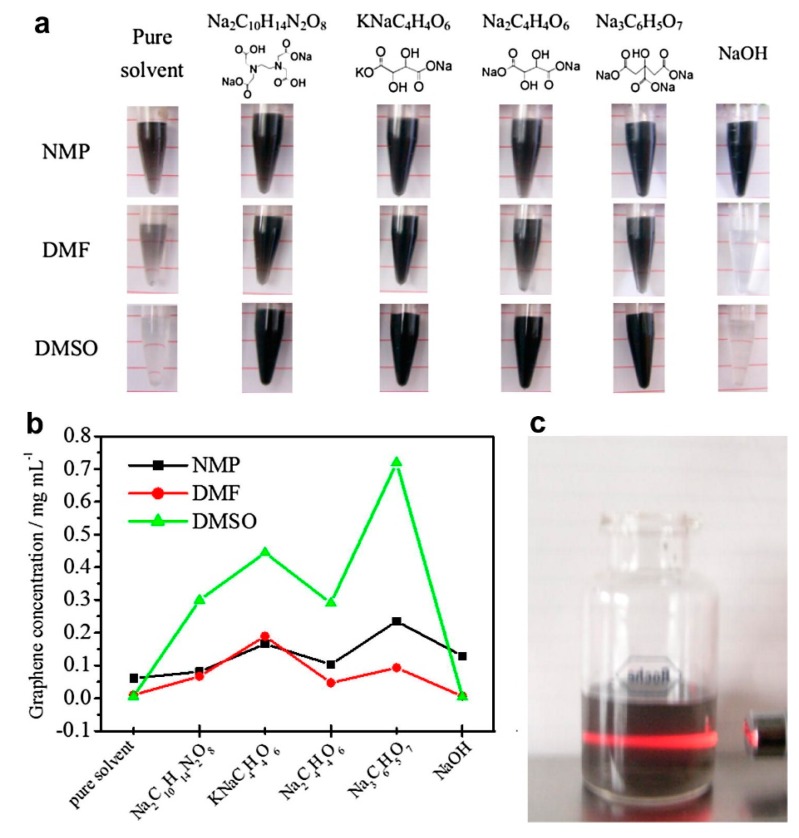
(**a**) Graphene sheets dispersed in organic solvents without or with various additives. (**b**) Concentrations of corresponding graphene dispersions in (**a**). (**c**) Tyndall effect exhibited by a diluted dispersion in DMSO passed through with red laser light. For interpretation of the references to color in this figure legend, the reader is referred to the web version of this article. Reproduced from [83]. Copyright Elsevier, 2013.

**Figure 17 nanomaterials-08-00942-f017:**
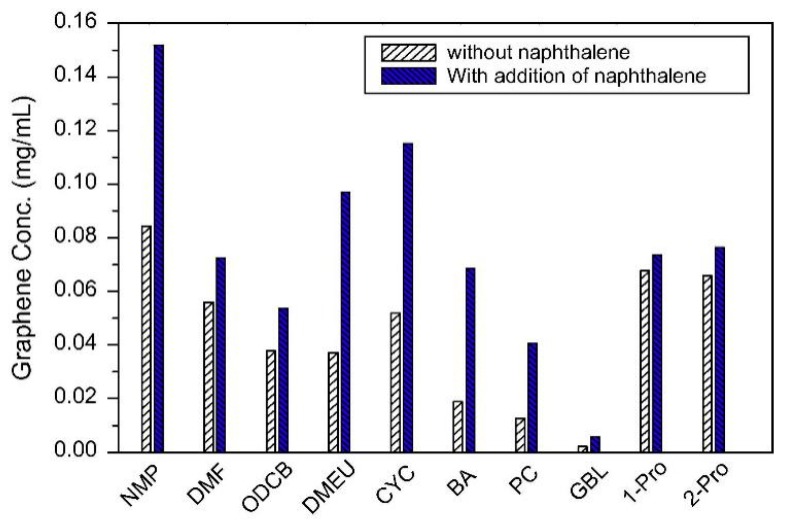
Concentrations of graphene sheets dispersed in organic solvents with or without addition of naphthalene. Reproduced from [84]. Copyright Elsevier, 2014.

**Figure 18 nanomaterials-08-00942-f018:**
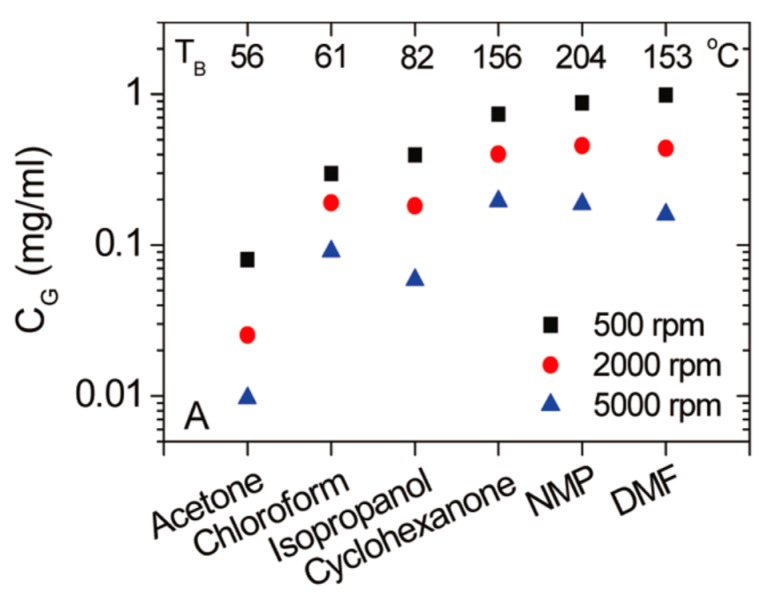
Concentration of dispersed graphene as a function of centrifugation rate for three high-boiling-point and three low-boiling-point solvents. The boiling points are given at the top of the figure. Reproduced from [84]. Copyright Elsevier, 2014.

**Figure 19 nanomaterials-08-00942-f019:**
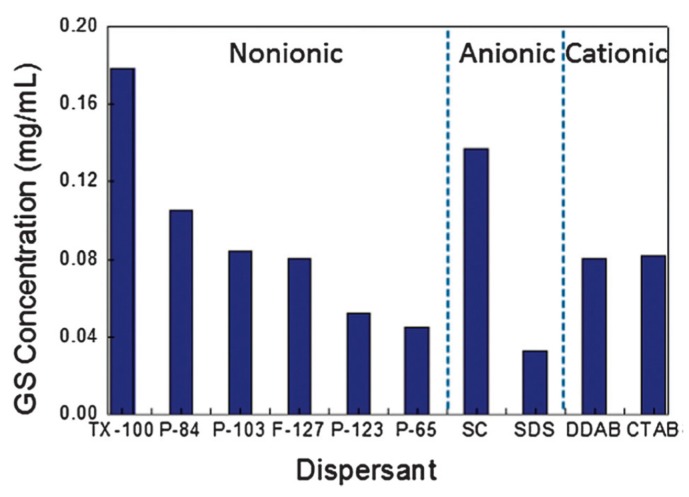
Graphene sheets concentration by tip–bath–tip (TBT) sonication for various dispersants, using identical TBT sonication treatment (integrated energy 23 kJ; 9 min tip sonication, 3 h bath sonication, and 9 min tip sonication). Reproduced from [50]. Copyright Owner Societies, 2013.

**Figure 20 nanomaterials-08-00942-f020:**
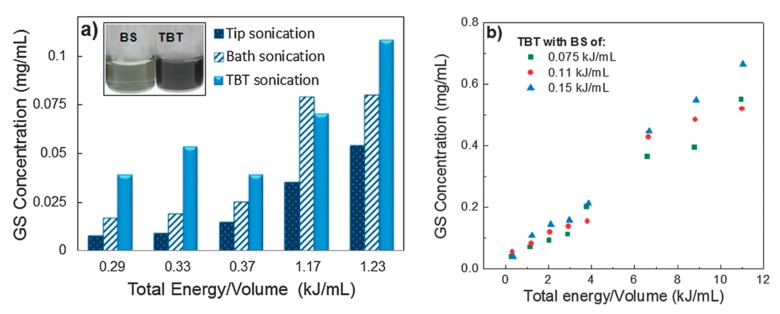
Optimization of sonication procedure, dispersant and energy/volume. (**a**) GS concentrations upon bath sonication (BS), tip sonication (TS) and tip-bath-tip sonication (TBT). Inset: image of the supernatant of the GS dispersions after centrifugation; (**b**) concentration of graphene sheets (with TX-100) as a function of integrated sonication energy/volume. These solutions were treated by TBT cycles. Reproduced from [50]. Copyright Owner Societies, 2013.

**Figure 21 nanomaterials-08-00942-f021:**
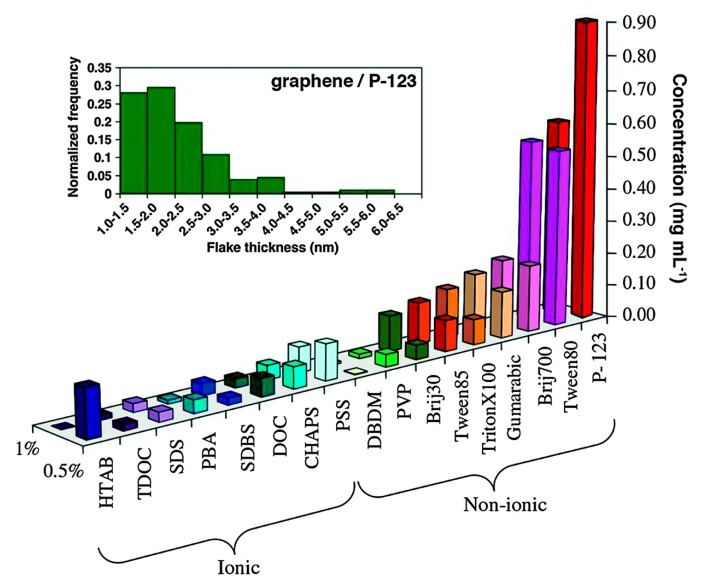
Concentration of graphene in aqueous dispersions achieved by the use of different surfactants, as estimated from UV–vis absorption measurements. Two surfactant concentrations are shown: 0.5% and 1.0% wt/vol. Inset figure-histogram showing the distribution of apparent flake thickness measured by AFM on 200 objects from dispersions stabilized by the non-ionic triblock copolymer P-123. Reproduced from [100]. Copyright Elsevier, 2011.

**Figure 22 nanomaterials-08-00942-f022:**
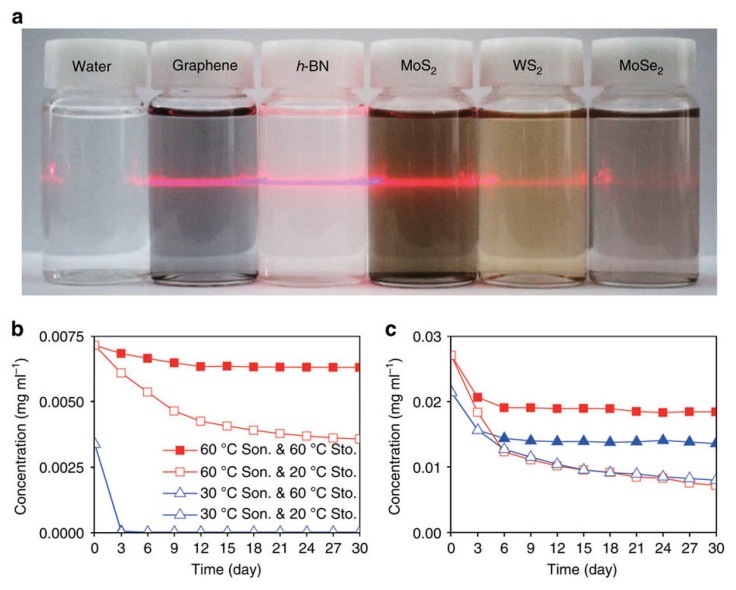
Temperature-dependent solution stabilities of the 2D materials in water. (**a**) Photographs of solutions of five 2D materials dispersed in deionized water for one month. The laser light across the solution bottles provides visual assistance because some suspended materials such as h-BN are not clearly visible. The long-term solution stabilities of (**b**) graphene, (**c**) h-BN sonicated at the high (60 °C) and low (30 °C) temperature and stored at high (60 °C) and low (20 °C) temperatures. In (**b**), two types of triangles almost overlap because of fast precipitation. In (**b**,**c**), squares and triangles denote high and low temperature sonication and solid and blank denote high and low temperature storage, respectively. Reproduced from [115]. Copyright Nature Publishing Group, 2015.

**Figure 23 nanomaterials-08-00942-f023:**
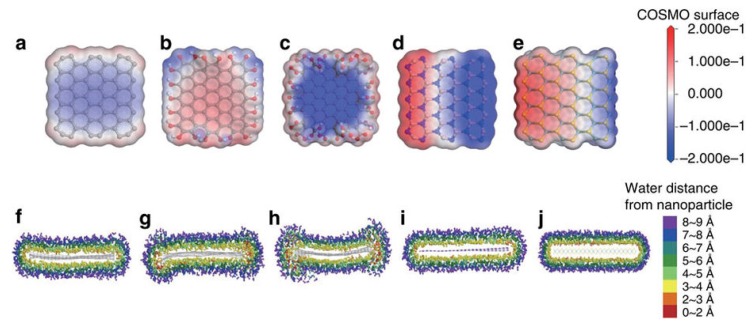
Molecular simulation results of solubility of 2D materials in water. Charge density of water around (**a**) pristine graphene, (**b**) –OH functionalized graphene, (**c**) –COOH functionalized graphene, (**d**) h-BN, and (**e**) MoS_2_. The isosurface is constructed from the COSMO-solvation model. The change of surface charge is much larger in polar nanoparticles. Water distributions around (**f**) pristine graphene, (**g**) –OH functionalized graphene, (**h**) –COOH functionalized graphene, (**i**) h-BN, and (**j**) MoS_2_. Reproduced from [115]. Copyright Nature Publishing Group, 2015.

**Figure 24 nanomaterials-08-00942-f024:**
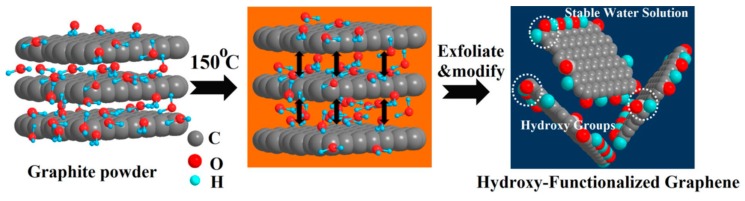
Schematic illustration of the exfoliation process. Reproduced from [117]. Copyright Nature Publishing Group, 2018.

**Figure 25 nanomaterials-08-00942-f025:**
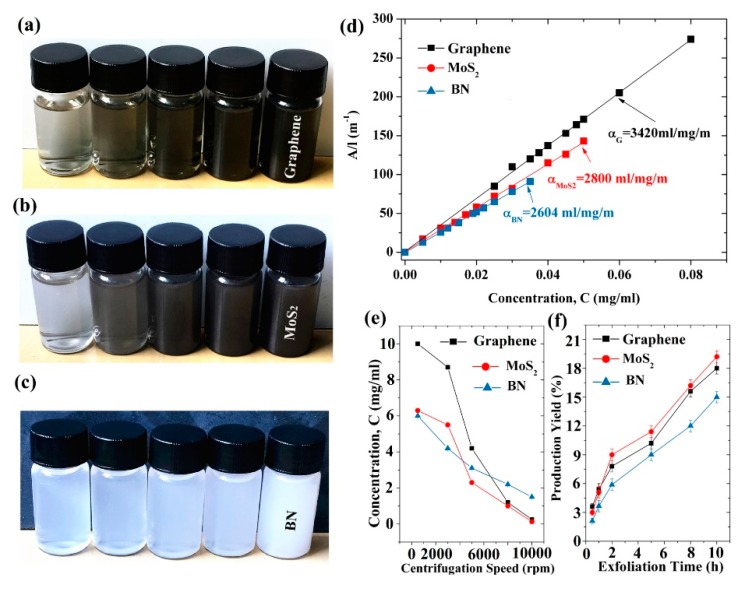
Photographs of aqueous dispersions of graphene (**a**), MoS_2_ (**b**), and BN (**c**) with different concentrations after centrifugation at 500–10,000 rpm for 30 min; (**d**) Lambert–Beer plots for graphene, MoS_2_, and BN; (**e**) dependence of 2D material concentration on centrifugation speed for the supernatant; (**f**) dependence of 2D material production yield on exfoliated time. Reproduced from [53]. Copyright IOP Publishing, 2017.

**Figure 26 nanomaterials-08-00942-f026:**
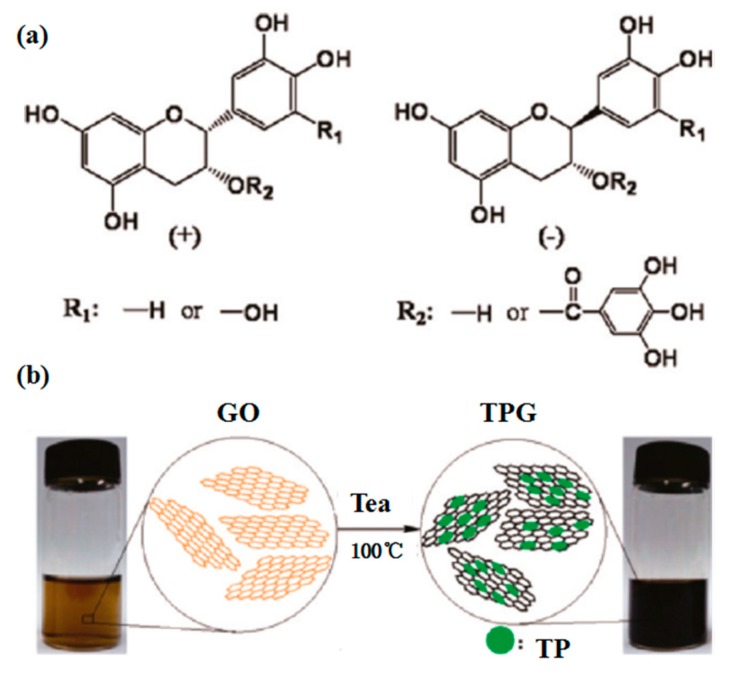
(**a**) Chemical structure of tea polyphenols (TPs). (**b**) Schematic illustration of the preparation of TP reduced graphene. Reproduced from [122]. Copyright American Chemical Society, 2011.

**Figure 27 nanomaterials-08-00942-f027:**
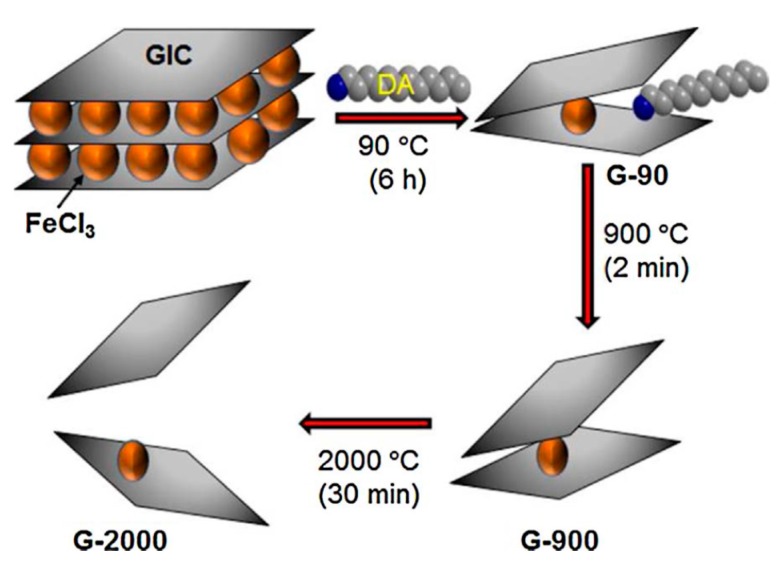
Schematic representation of the production of G-2000. GIC: graphite intercalated compound; DA: dodecylamine. Reproduced from [124]. Copyright Elsevier, 2018.

**Figure 28 nanomaterials-08-00942-f028:**
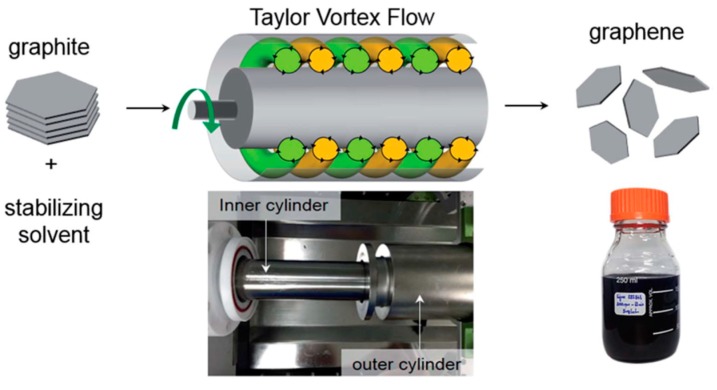
Schematic of the shear-exfoliation of graphite into few-layer graphene by a Taylor Vortex flow. Photographs shows a Taylor-Couette flow reactor and the graphene dispersions produced by shear exfoliation. Reproduced from [55]. Copyright Royal Society of Chemistry, 2016.

**Figure 29 nanomaterials-08-00942-f029:**
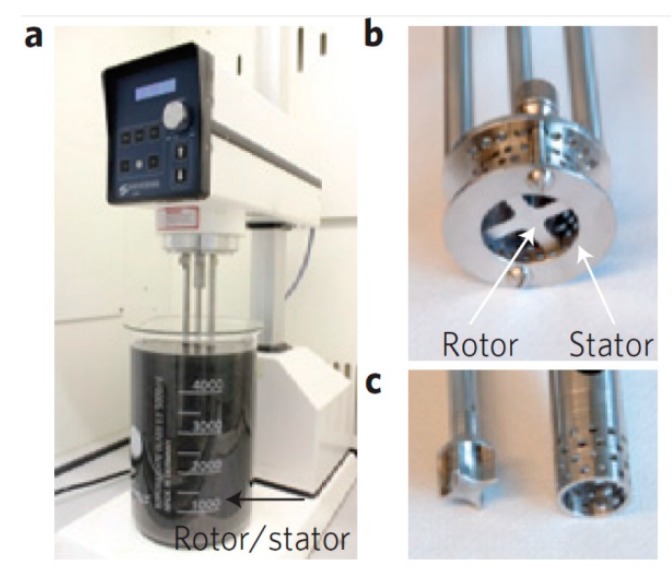
(**a**) A Silverson model L5M high-shear mixer with mixing head in a 5 L beaker of graphene dispersion. Close-up view of a D = 32 mm mixing head (**b**) and D = 16 mm mixing head with rotor (left) separated from stator (**c**) Reproduced from [19]. Copyright Nature Publishing Group, 2014.

**Figure 30 nanomaterials-08-00942-f030:**
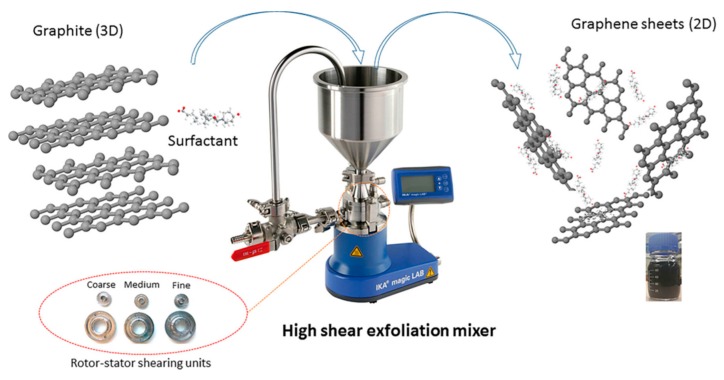
Schematic illustration of the exfoliation process using a high shear colloidal mixer. Reproduced from [24]. Copyright Springer, 2017.

**Figure 31 nanomaterials-08-00942-f031:**
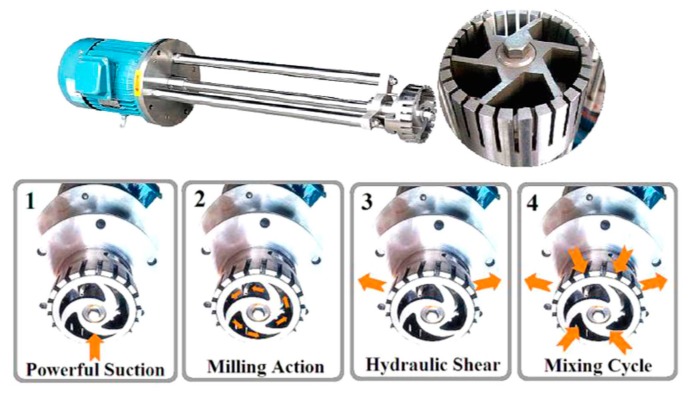
FJ300-S model high-shear mixer and mixer operation. Reproduced from [53]. Copyright IOP Publishing, 2017.

**Figure 32 nanomaterials-08-00942-f032:**
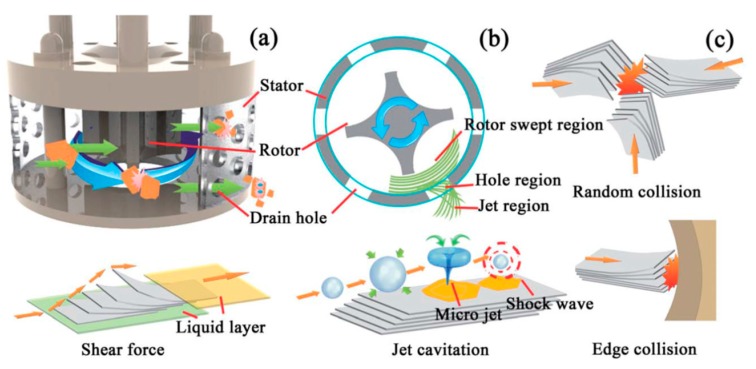
(**a**) 3D sectional drawing of the high shear generator; (**b**) main energy dissipation regions of the high shear mixer (sectional view); (**c**) the schematic model of preparing GNSs by shear force, collision, and jet cavitation. Reproduced from [56]. Copyright Royal Society of Chemistry, 2014.

**Figure 33 nanomaterials-08-00942-f033:**
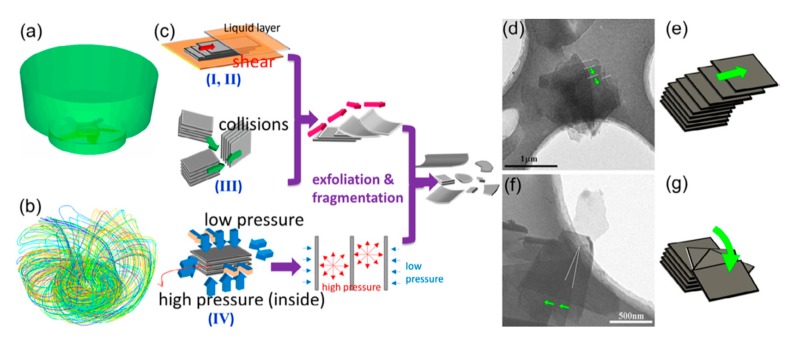
(**a**) Schematic of the simplified model used for computational fluid dynamics analyses. (**b**) Pathlines released form the impeller surface colored by particle ID. (**c**) Illustration for the exfoliation mechanism. Deliberately captured partially exfoliated FLG flakes with translational (**d**,**e**) and rotational (**f**,**g**) lateral exfoliation. Reproduced from [127]. Copyright Elsevier, 2014.

**Table 1 nanomaterials-08-00942-t001:** Graphene preparation process of some Chinese companies.

No	Preparation Methods	Companies
1	Reduction of Graphene Oxide	Ningbo Morch Technology Co., Ltd. (Ningbo, China) The Sixth Element (Changzhou) Materials Technology Co., Ltd. (Changzhou, China) SUNANO ENERGY (Taizhou, China) Shanghai Carbon Source Huigu New Material Technology Co., Ltd. (Shanghai, China) Jining Lite Nanotechnology Co., Ltd. (Jining, China) Beijing Carbon Century Technology Co., Ltd. (Beijing, China) Xiamen Kaina Graphene Technology Co., Ltd. (Xiamen, China)
2	Liquid-Phase Exfoliation	Deyang Carbonene Technology Co., Ltd. (Deyang, China) Hengli Sheng Tai (Xiamen) Graphene Technology Co., Ltd. (Xiamen, China) Shanghai LEVSON Nanotechnology Co., Ltd. (Shanghai, China)
3	Supercritical Fluid Exfoliation	Nanjing Ke Fu Nanotechnology Co., Ltd. (Nanjing, China) Qingdao Delta Nanotechnology Co., Ltd. (Qingdao, China)
4	Chemical Vapor Deposition	Chongqing Mo Xi Technology Co., Ltd. (Chongqing, China) Changzhou Two Dimensional Carbon Technology Co., Ltd. (Changzhou, China)

**Table 2 nanomaterials-08-00942-t002:** Concentration of graphene produced via sonication in relation to exfoliation time and media.

No	Method	Exfoliation Medium	Time (h)	Concentration (mg mL^−1^)	Reference
1	Bath sonication	NMP	0.5	0.01	[21]
2	Sonication	NMP	460	1.2	[40]
3	Bath sonication	O-DCB	8	0.0066	[43]
4	Sonication (sequential)	NMP	8	0.43	[44]
5	Tip sonication	NMP/azobenzene	3	0.07	[45]
6	Bath sonication	NMP/NaOH	1.5	0.07	[46]
7	Bath sonication	Water/SC	430	0.3	[34]
8	Sonication	Water/SC	12	0.25	[47]
9	Sonication	Water/PVP	1	0.42	[48]
10	Bath sonication	Water/ammonia solution	8	0.058	[49]
11	Sonication	Water/Triton X-100	12	0.7	[50]
12	Sonication	Low boiling point solvents	48	0.5	[51]

**Table 3 nanomaterials-08-00942-t003:** Concentration of graphene produced via high-shear mixing in various solvents.

No	Exfoliation Medium	Exfoliation Time (h)	Concentration (mg mL^−1^)	Reference
1	NMP	0.5	0.01	[19]
2	Water/SC	2	1.1	[24]
3	Water/PVP	2	0.7	[24]
4	Water/Black liquor	10	10	[53]
5	Water/Black tea	0.25	0.032	[54]
6	NMP	1	0.65	[55]
7	40 vol % IPA–water	1	0.27	[56]
8	NMP	1	1	[57]
9	NMP	6	0.251	[58]
10	OCDB	1.5	0.03	[59]

**Table 4 nanomaterials-08-00942-t004:** Original organic solvents have been used to exfoliate graphene.

No	Organic Solvents	Surface Tension (mJ m^−2^)	Boiling Point (°C)	Chemical Structure	Concentration (mg mL^−1^)	Reference
1	*N*-methyl-2-pyrrolidone (NMP)	40	203	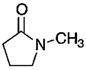	0.01	[21,77]
2	γ-butyrolactone (GBL)	46.5	204	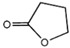	<0.01	[21]
3	*N,N*-dimethylacetamide (DMAC)	36.7	165	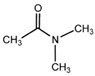	<0.01	[21]
4	*N,N*-dimethylformamide (DMF)	37.1	154	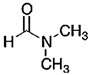	0.0041~0.0045	[78,79]
5	Dimethylsulfoxide (DMSO)	42.9	189	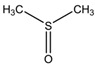	0.0037~0.0041	[78,79]
6	Ortho-dichlorobenzene (ODCB)	37	181	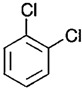	0.03	[59]

**Table 5 nanomaterials-08-00942-t005:** Production of aqueous graphene dispersions using different surfactants and addition methods.

Surfactant	Type of Surfactant	Concentration to Achieve γ = 41 mJ m^−2^	Batch Concentration (mg mL^−1^)	Continuous Concentration (mg mL^−1^)
Pluronic^®^108	Nonionic	0.1%	0.11	10.23
Pluronic^®^127	Nonionic	0.1%	0.078	6.55
Sodium dodecylsulfate (SDS)	Anionic	7 mM	0.06	4.92
Hexadecyltrimethylammonium bromide (CTAB)	Cationic	0.6 mM	0.05	4.05
Tetradecyltrimethylammonium bromide (TTAB)	Cationic	2.1 mM	0.055	5.01
Dodecyltrimethylammonium bromide (DTAB)	Cationic	10 mM	0.06	5.22

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
