# Peer review of "Liquid-Phase Exfoliation of Graphene: An Overview on Exfoliation Media, Techniques, and Challenges"

_nanomaterials, 2018, doi:10.3390/nano8110942_

Reviewer 1 Report

Authors summarize the exfoliation techniques for preparation of 2D materials (mainly graphene). 2D materials are the key parts to resolve the energy issue and exfoliation route is one of the promising candidate for preparation of 2D materials. The review is well-organized and very helpful for developing the field. I will use the review for my future work as a reference. However, there are some small mistakes about font, superior, inferior and so on. I hope that they be revised in proof.

Author Response

We are grateful to the reviewer for the useful and constructive comments and are very pleased to observe that the reviewer is positive for this paper and recommends its acceptance, provided that his/her comments can be addressed. We are happy to address these comments.

Point 1: Authors summarize the exfoliation techniques for preparation of 2D materials (mainly graphene). 2D materials are the key parts to resolve the energy issue and exfoliation route is one of the promising candidate for preparation of 2D materials. The review is well-organized and very helpful for developing the field. I will use the review for my future work as a reference. However, there are some small mistakes about font, superior, inferior and so on. I hope that they be revised in proof.

Response 1: We are very sorry for our negligence of some mistakes about font, superior and inferior. According to the reviewer’s comment, we have changed the font in the paper. The font of “Palatino Linotype” is adopted and the font size is adjusted according to the requirement of Nanomaterials. We have also corrected some other formatting errors and grammatical mistakes. Furthermore, we have had the manuscript polished with a native English speaking colleague.

Thank you very much for your comments and suggestions.

Reviewer 2 Report

This paper reviews the latest progress on graphene reparation in various media, which includes organic solvents, ionic liquids, water/polymer etc. The paper is well written, and the authors include the necessary bibliography citations needed. The figures are very helpful and provide important information needed for this work. The paper should be accepted under minor revisions shown below:

 1) Replace “Tab.” with “Table” in all table captions. Moreover, “Figure X” in the figure captions should be “Figure X.”, where X = 1, 2, 3, etc. (i.e., insert the period after number “X”).

2) The font in the paper is not uniform and needs to be changed.

3) “ref x” appears in red font and needs to be in black font.

4) Roman-type numerals shown as subscripts and appear in the bibliography needs to be removed.

Author Response

We are grateful to the reviewer for the useful and constructive comments and are very pleased to observe that the reviewer is positive for this paper and recommends its acceptance, provided that his/her comments can be addressed. We are happy to address these comments.

Point 1: Replace with in all table captions. Moreover, “Figure X” in the figure captions should be “Figure X.”, where X = 1, 2, 3, etc. (i.e., insert the period after number “X”). 

Response 1: It is our negligence and we are sorry about this. We have replaced “Tab.” with “Table” in all table captions.  We have also replaced “Figure X” with “Figure X.” in all figure captions.

Point 2: The font in the paper is not uniform and needs to be changed.

Response 2: According to the reviewer’s comment, we have changed the font in the paper. The font of “Palatino Linotype” is adopted and the font size is adjusted according to the requirement of Nanomaterials.

Point 3: “ref x” appears in red font and needs to be in black font.

Response 3: As pointed out by the reviewer, we have corrected this mistake, and we have replaced Adapted from ref. 1 with permission from the Nature Publishing Group. with “Reproduced from [1]. Copyright Nature Publishing Group, 2007.” according to the requirement of Nanomaterials.

Point 4: Roman-type numerals shown as subscripts and appear in the bibliography needs to be removed.

Response 4: According to the reviewers comment, we have removed Roman-type numerals shown as subscripts. And we have adjusted notation of references in revised manuscript.

Thank you very much for your comments and suggestions.

Reviewer 3 Report

Even though, as also stated by the authors, there are quite a few review articles in literature about LPE of graphene, the review by Xu et al. is a very nice work, presenting a systematic study of all possible ways for getting graphene with LPE, the media and infrastructure used. I suggest publication upon minor revision. I have the following comments:

As rational for review articles where many people write some parts and then these are combined, there are some inconsistencies in the format used and some parts may have 2-3 more grammar mistakes or replicate discussion. So, somebody has to eliminate these.

You must use Arabic numbers for references in the main text.

In section 4 you mention also microfluidization but there is not a relevant subsection as for the rest mentioned techniques.

Author Response

We are grateful to the reviewer for the useful and constructive comments and are very pleased to observe that the reviewer is positive for this paper and recommends its acceptance, provided that his/her comments can be addressed. We are happy to address these comments.

Point 1: As rational for review articles where many people write some parts and then these are combined, there are some inconsistencies in the format used and some parts may have 2-3 more grammar mistakes or replicate discussion. So, somebody has to eliminate these.

Response 1: According to the reviewer’s comment, we have revised the whole manuscript carefully and tried to avoid grammatical mistakes. In addition, we have sought the assistance of a native English-speaking colleague to check the manuscript and refined the language carefully.

Point 2: You must use Arabic numbers for references in the main text.

Response 2: It is our negligence and we are sorry about this. As pointed out by the reviewer, Arabic numbers for references have been used in the main text.

Point 3: In section 4 you mention also microfluidization but there is not a relevant subsection as for the rest mentioned techniques.

Response 3: Thank you for your valuable advice. In Section 2, the techniques of liquid-phase exfoliation, i.e. sonication, high-shear mixing and microfluidization, we have given a brief overview on microfluidization. Compared with sonication and high-shear mixing, there are less research studies on microfluidization that is used to prepare graphene by liquid-phase exfoliation. Thus, microfluidizer, a high-pressure homogenizer, offering high shear, has been not introduced in detail in section 4.

Thank you very much for your comments and suggestions.

Round  2

Reviewer 2 Report

The manuscript can be now accepted as is.

Reviewer 3 Report

The revised manuscript is very nice. Authors have responded to my comments and I agree with them that there are not many works on microfluidazation hence no need for detailed description in section 4 since there is a brief one at 2.3.